# Does Diabetes Mellitus Increase the Risk of Avascular Osteonecrosis? A Systematic Review and Meta-Analysis

**DOI:** 10.3390/ijerph192215219

**Published:** 2022-11-18

**Authors:** Wojciech Konarski, Tomasz Poboży, Andrzej Kotela, Andrzej Śliwczyński, Ireneusz Kotela, Martyna Hordowicz, Jan Krakowiak

**Affiliations:** 1Department of Orthopaedic Surgery, Ciechanów Hospital, 06-400 Ciechanów, Poland; 2Faculty of Medicine, Collegium Medicum, Cardinal Stefan Wyszynski University in Warsaw, Woycickiego 1/3, 01-938 Warsaw, Poland; 3Social Medicine Institute, (Department of Social and Preventive Medicine), Medical University of Lodz, 90-419 Lodz, Poland; 4Department of Orthopedic Surgery and Traumatology, Central Research Hospital of Ministry of Interior, Wołoska 137, 02-507 Warsaw, Poland; 5General Psychiatry Unit III, Dr. Barbara Borzym’s Independent Public Regional Psychiatric Health Care Center, 26-600 Radom, Poland

**Keywords:** avascular osteonecrosis, bone necrosis, diabetes mellitus, AVN

## Abstract

Avascular osteonecrosis (AVN) is caused by the disrupted blood supply to the bone. Most AVN cases occur in the femoral head, but other sites might be affected as well, including the jaw or distal bones of the extremities. Previous studies suggested that diabetes could increase the risk of AVN of the jaw, but the relationship between diabetes and AVN in other bone sites is unclear. This systematic review and meta-analysis aimed to summarize the evidence from studies that had reported on the occurrence of AVN in sites other than the jaw, depending on the diagnosis of diabetes. Overall, we included 6 observational studies carried out in different populations: primary or secondary AVN of the femoral head, Takayasu arteritis, general population, kidney transplant recipients, systemic lupus erythematosus, and primary brain tumors. A random-effects meta-analysis showed that the risk of AVN in sites other than the jaw was non-significantly increased in patients with diabetes (odds ratio: 1.90, 95% confidence interval: 0.93–3.91). The pooled estimate increased and was significant after the exclusion of one study (2.46, 1.14–5.32). There was a significant heterogeneity (I^2^ = 65%, tau^2^ = 0.48, *p* = 0.01; prediction interval, 0.21–16.84). There was no significant publication bias (*p* = 0.432). In conclusion, diabetes could increase the risk of AVN in sites other than the jaw, but the available evidence is limited. There is a need for large, well-designed, population-based studies.

## 1. Introduction

Avascular osteonecrosis (AVN) refers to the death of bone cells due to disruption of the blood supply, which typically occurs in bone areas with limited collateral circulation [1]. The death of bone cells is followed by the disappearance of the articular surface and subsequent degenerative arthritis, which may lead to progressive disability [2,3,4]. Moreover, AVN may increase the risk of tumorigenesis in the affected sites in the long-term [5]. AVN occurs in various bones, such as the femoral head, jaw, knee, talus, or humerus [6,7,8]. The femoral head is the most common site of AVN, with an estimated incidence in the United States between 10,000 and 20,000 cases per year [9]. Femoral-head AVN is diagnosed mostly in young adults (mean age of onset ~40 years) [10] and is associated with a substantial disease burden [11]. Magnetic resonance imaging (MRI) is a highly effective method for the early detection and staging of the disease [12]. AVN of the jaw is a debilitating disease that occurs predominantly in people using bisphosphonates, which are medications that inhibit bone resorption [13,14]. The treatment of early-stage AVN primarily involves physiotherapy, whereas late-stage disease often requires surgery [15,16,17].

AVN may be caused by trauma, such as fracture, particularly displaced fractures [18], or it can be non-traumatic. The pathogenesis of non-traumatic osteonecrosis is unclear, but there are several established risk factors such as corticosteroid treatment, alcohol abuse, rheumatic diseases, bone-marrow transplantation, or antiretroviral treatment [15,19,20,21,22,23]. Diabetes mellitus (DM), which causes disease of both large vessels (macroangiopathy) and small vessels (microangiopathy) [24], can also increase the risk of avascular osteonecrosis [25,26,27]. Various pathogenic mechanisms in diabetes might additionally impair bone formation, accelerate the apoptosis of bone cells, and reduce the formation of new micro-vessels [1]. Figure 1 presents a case of bilateral AVN in a 62-year-old patient with a history of DM with no other comorbidities.

Previous studies on medication-related osteonecrosis of the jaw reported that 15–70% of patients with this condition had diabetes [28,29]. In contrast, the evidence on the association between diabetes and the risk of AVN in sites other that the jaw is less clear. Therefore, we conducted a systematic review and a meta-analysis of studies that had compared cohorts of patients with or without AVN in sites other that the jaw, reporting the proportions of patients with and without diabetes.

## 2. Materials and Methods

### 2.1. Protocol and Registration

The protocol for this systematic review and meta-analysis was prepared in compliance with the preferred reporting items for systematic review and meta-analysis protocols (PRISMA) guidance [30]. The protocol is available as Appendix A.

### 2.2. Search Strategy

We searched the PubMed and EMBASE databases for studies published until 29 July 2022 in English, Polish, or Spanish. We used the following keywords: “AVN”, “sterile necrosis”, “osteonecrosis”, “diabetes”, “hyperglycemia”, and “risk factor”. The exact search phrases are shown in the study protocol. The titles and abstracts were screened to select studies for full-text review. When studies were carried out in overlapping samples, we chose the one with the greatest sample size.

### 2.3. Study Selection

Two investigators assessed the eligibility of the studies. The more experienced investigator decided on the eligibility when there was a discrepancy. Included were studies that reported sufficient data to obtain or calculate the total number of cases and the number of AVN events in groups with diabetes and without diabetes. Eligible were studies with a score of 5 or more on the Newcastle–Ottawa Scale to exclude studies with a high risk of bias (scores 4 or lower). Case reports, case series among 15 patients or fewer, reviews, and letters to the editor were excluded.

### 2.4. Data Extraction and Quality Assessment

Two investigators independently extracted the following data: the name of the first author, the publication year, the study design, and characteristics of AVN cases (corticosteroid use, the mean age, and the proportion of men).

### 2.5. Data Synthesis and Analysis

We calculated odds ratios (OR) using inverse variance for the risk of AVN in patients with diabetes vs. without diabetes (OR > 1 indicated an increased risk of AVN in patients with diabetes). A random-effects meta-analysis was carried out with the restricted maximum-likelihood estimator for tau^2^ and the Q-profile method for the confidence interval of tau^2^ and tau. Heterogeneity was expressed with the I^2^ and τ^2^ statistics, and it was evaluated with Cochran’s Q test. A prediction interval was estimated to take heterogeneity into account. Sensitivity analyses included leave-one-out analyses. Publication bias was assessed using the Eggers regression test and by inspection of a funnel plot and using fail-safe N analysis. A *p*-value of less than 0.05 was considered statistically significant. The R software (version 4.1.3) was used for all analyses.

## 3. Results

A total of 963 citations were identified, and 39 articles were retrieved in full text. After a full-text review, 31 studies were excluded due to insufficient data, 1 due to poor quality (low NOS score) and 1 due to an overlapping cohort (Figure 2). Overall, 6 studies were included [31,32,33,34,35,36].

The studies were carried out in various populations: primary or secondary AVN of the femoral head, Takayasu arteritis, general population, kidney transplant recipients, systemic lupus erythematosus, and primary brain tumors. Nearly all of the AVN cases were reported in the femoral head, with few cases in the femoral condyle, humerus, and ankle (see Table 1). The mean age among AVN cases ranged from 34 to 57 years, and the proportion of males ranged from 13% to 100%. Corticosteroid use was reported in a substantial proportion of AVN cases. The detailed characteristics of included studies are given in Table 1.

The ORs for AVN risk among patients with diabetes in individual studies ranged from 0.81 to 8.02. The ORs among patients with diabetes were significantly increased in three studies: Lai et al. [33], Tse et al. [35], and Lim et al. [36], whereas the relationship between diabetes and the risk of osteonecrosis was non-significant in the remaining studies (see Figure 3 for details).

The pooled estimate indicated a non-significantly increased risk of AVN in patients with diabetes (OR = 1.90, 95%CI: 0.93–3.91; Figure 3). There was significant heterogeneity (I^2^ = 65%, tau^2^ = 0.48, and *p* = 0.01; prediction interval, 0.21–16.84; Figure 3). There was no asymmetry in the funnel plot (Figure 4). There was no significant publication bias (Egger’s regression: t = 1.0807, df = 4, and *p* = 0.34). Based on Rosenthal’s analysis, the number of fail-safe (*n* = 15) was greater than the number of included observations (*n* = 6) (*p* = 0.001), which means there is no risk of publication bias.

The pooled estimate did not change substantially in leave-one-out analyses, but the estimate was significant after excluding the Yang et al. study (OR = 2.46, 95%CI: 1.14–5.32; Figure 4). Heterogeneity remained substantial after the exclusion of any of the studies (I^2^ of 58% or greater, Figure 5).

## 4. Discussion

This systematic review and meta-analysis summarized the results from observational studies carried out among patients with AVN in bone sites other than the jaw depending on the diagnosis of diabetes. Overall, the available evidence is scarce, with four of the sic studies reporting eight or fewer AVN cases among patients with diabetes. Of the two large studies, the population-based study by Lai et al., designed specifically to analyze diabetes as a risk factor of AVN, reported a significantly increased risk of femoral head AVN in patients with diabetes [33]. The other large study (Yang et al. 2020 [31]), carried out in a single orthopedic hospital found a non-significantly lower risk of AVN in patients with diabetes. The pooled estimate in our meta-analysis indicated an increased risk of AVN in bone sties other than the jaw among patients with diabetes, but the effect was non-significant. Moreover, a substantial heterogeneity resulted in a very wide prediction interval. We found no significant publication bias.

MRI was used in all studies to diagnose AVN. Only the study based on the Taiwanese registry diagnosis was made based on plain film, nuclear scan, or MRI. MRI is useful in the early diagnosis of AVN and can identify patients at risk of femoral head fracture [15].

Lai et al. published the largest study to date, with 146 cases of non-traumatic AVN of the femoral head among patients with diabetes [33]. In that nationwide study from Taiwan, a cohort with diabetes was compared with a cohort without diabetes matched for sex, age, corticosteroid use, and several comorbidities; the risk of avascular osteonecrosis was significantly greater in patients with diabetes (hazard ratio = 1.16; 95%CI, 1.11–1.21) [33]. The second-largest study in our systematic review was the case-control study by Yang et al. [31], who compared patients with femoral head osteonecrosis (*n* = 755) and those with fractures of extremities (*n* = 489). These investigators reported that diabetes was associated with a non-significantly lower risk of femoral head AVN (OR = 0.81; 95%CI, 0.52–1.26). However, the study was carried out in a single hospital, and the control group was older than the patients with AVN (65 vs. 57 years). This age imbalance could partly account for the largest prevalence of diabetes in the control group than the AVN group (9.2% vs. 7.6%), which resulted in a lower risk of AVN associated with diabetes. After the exclusion of the study by Yang et al., the pooled estimate increased and was significant (OR = 2.46, 95%CI, 1.14–5.32). Felten et al. studied the risk of AVN in 805 kidney transplant recipients, with a total of 18 cases of new-onset avascular necrosis (8 in patients with diabetes); patients with or without new-onset AVN after transplant were of a similar age (51 vs. 55 years) [34]. The significant risk factors of AVN included pre-transplant diabetes, high corticosteroid doses, obesity, and hyperparathyroidism [34]. Tse and Mok reported 55 cases of AVN among 277 patients with systemic lupus erythematosus; although the risk for AVN was significantly increased in those with diabetes, there were only 4 cases of AVN in these patients [35]. Similarly, in the study by Lim et al., among 144 patients with primary brain tumors who received corticosteroids, the risk of AVN was increased significantly, but there were only 9 patients with diabetes [36]. The risk of AVN was not related to diabetes in the study by Gokcen et al. among 29 patients with Takayasu arteritis, but all of the patients in that study used corticosteroids [32].

A potential relationship between the risk of AVN and diabetes was first noticed among patients treated with bisphosphonates who developed AVN of the jaw. Bisphosphonates, which inhibit osteoclastic bone resorption, are used commonly to treat bone diseases such as osteoporosis, multiple myeloma, and bone metastases [37]. A systematic review of 27 studies estimated that 2.7% of bisphosphonate users develop AVN of the jaw [38]. Although it is unclear how bisphosphonates cause jaw necrosis, various mechanisms are suspected, including decreased bone remodeling, impaired wound healing, and local anti-angiogenic effects [39]. Several studies reported that the risk of jaw AVN might be increased by diabetes, which may inhibit bone formation, increase apoptosis of bone cells, and impair angiogenesis [28]. However, there was no significant relationship between diabetes and the risk of AVN of the jaw in a population-based study among 900 patients (largest to date) [40]. Our systematic review and meta-analysis suggest that the association between diabetes and the risk of AVN in bones other that the jaw is also debatable.

Our study had several limitations. We included only six studies, most of which were not designed specifically to study diabetes as a risk factor for AVN in sites other than the jaw. Moreover, the generalizability of the meta-analysis results is limited by substantial heterogeneity, which could be due to the inclusion of studies carried out in very heterogenous populations. The heterogeneity did not decrease in the leave-one-out sensitivity analyses, and there were too few studies to carry out subgroup analyses for different populations. Moreover, corticosteroid use, an established risk factor for AVN, was prevalent in the studies included.

## 5. Conclusions

In conclusion, although the pooled estimate from this meta-analysis suggests that diabetes might increase the risk of AVN in sites other than the jaw, the existing evidence is limited. The relationship between diabetes and the risk AVN should be further investigated in well-designed, population-based studies with well-matched control groups.

## Figures and Tables

**Figure 1 ijerph-19-15219-f001:**
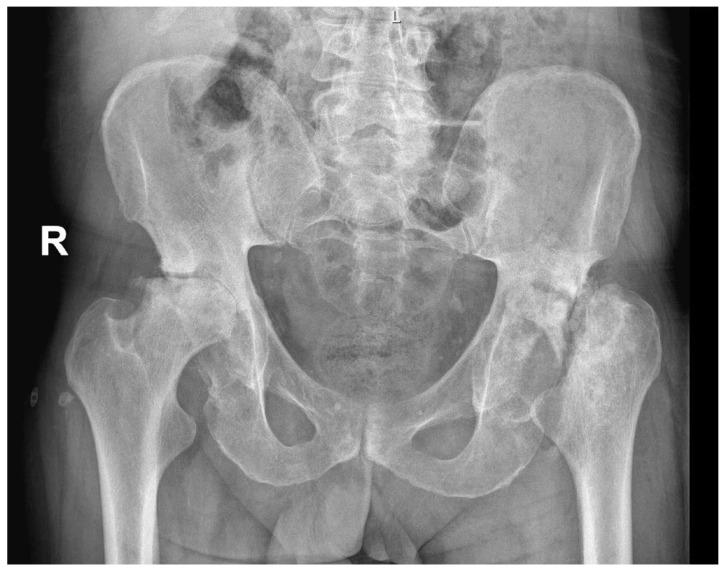
Radiological image of bilateral avascular osteonecrosis in a patient with diabetes mellitus with no other comorbidities. R—right.

**Figure 2 ijerph-19-15219-f002:**
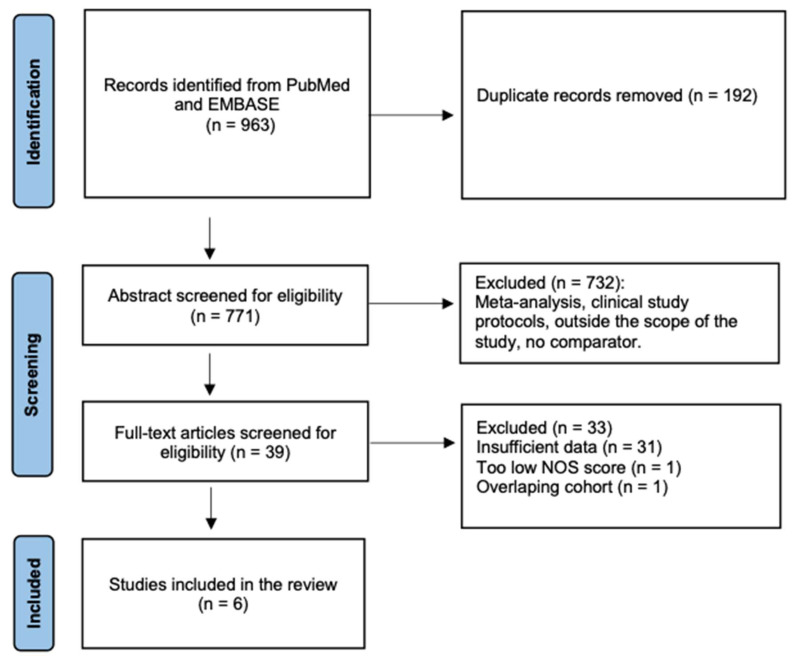
Flow chart of the study selection.

**Figure 3 ijerph-19-15219-f003:**
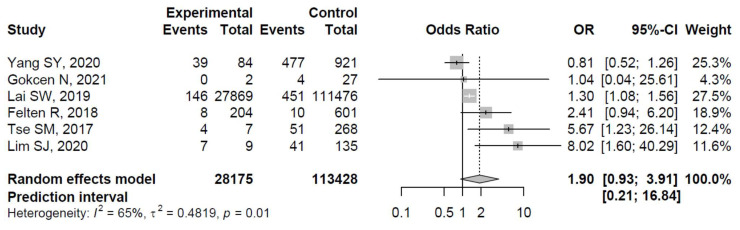
Forest plot showing the association between diabetes and risk of osteonecrosis in sites other than the jaw (odds ratios > 1 indicate increased risk of AVN in sites other than the jaw among patients with diabetes) [31,32,33,34,35,36].

**Figure 4 ijerph-19-15219-f004:**
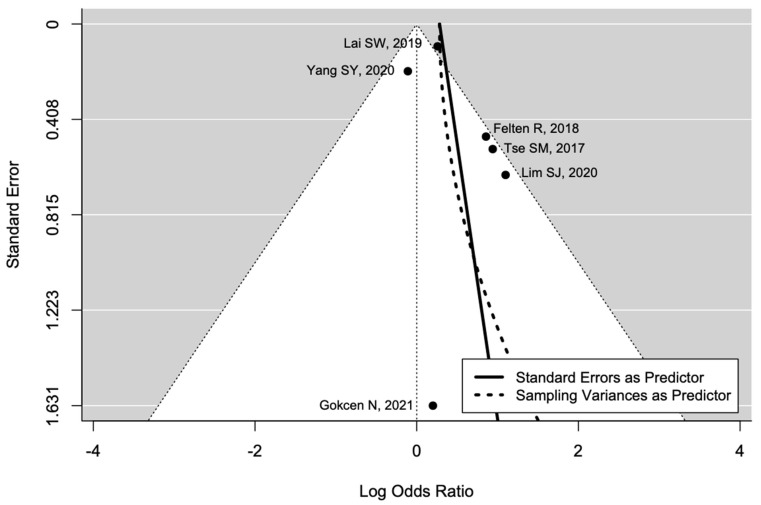
Funnel plot for publication bias [31,32,33,34,35,36].

**Figure 5 ijerph-19-15219-f005:**
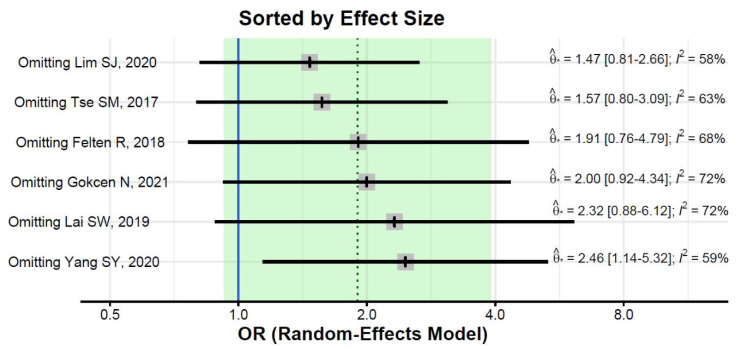
Leave-one-out sensitivity analyses [31,32,33,34,35,36].

**Table 1 ijerph-19-15219-t001:** Characteristics of individual studies.

Study	Population	Study Design	AVN Sites	Mean Age in AVN Cases	Male % in AVN Cases	Corticosteroid Use in AVN Cases	NOS
Yang SY, 2020 [31]	Primary and secondary femoral head AVN vs. extremity factures	Retrospective, case-control	Femoral head	57	52%		6
Gokcen N, 2021 [32]	Takayasu arteritis	Retrospective cross sectional	Femoral head	38	100%	Mean duration of corticosteroid use of 7.5 years	6
Lai SW, 2019 [33]	General population (diabetes vs. non-diabetes)	Retrospective cohort study	Femoral head	-	-	-	9
Felten R, 2018 [34]	Kidney transplant recipient	Retrospective cohort study	Femoral head (15); femoral condyle (2); and ankle (1)	55	22%	Median total dose of prednisolone 3 month after transplant was 2594 mg	8
Tse SM, 2017 [35]	Systemic lupus erythematosus	Retrospective cohort study	Hip (82%); femoral condyle (9%); and humeral head (5%)	34	13%	All used corticosteroids	8
Lim SJ, 2020 [36]	Primary brain tumors after surgery	Retrospective cohort study	Femoral head	36	50%	All used corticosteroids	9

NOS, Newcastle–Ottawa Scale.

## Data Availability

Data are available from the corresponding author upon reasonable request.

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
