# Peer review of "Does Diabetes Mellitus Increase the Risk of Avascular Osteonecrosis? A Systematic Review and Meta-Analysis"

_ijerph, 2022, doi:10.3390/ijerph192215219_

Round 1

Reviewer 1 Report

The study is well performed and although there is a limited amount of studies to be pooled. The overall effect is sound. 

The authors have provided a study of the heterogeneity (given the limitation of the study number). While I really like the way the sensitivity analysis is presented maybe a more detailed analysis of the funnel plot through the Egger test would be interesting. Moreover, the Rosenthal fail save number may help the reader understand the possibility that any missing study would change the observed effect. 

Author Response

Dear Reviewer.

Thank you for this review. According to the suggestion, we have performed Egger’s test and Rosenthal's analysis. Also, we have replaced figure 4 with a new one. 

Best redargs

Reviewer 2 Report

1. The value of MRI in early detection of AVN is missing. 

2. The legend of figure 1 is inadequate. The left hip shows destructive osteoarthritis which, if the patient had AVN, is the result of insufficiency fracture and subsequent bone resorption.

3. It is not clear if the studies included in the analysis applied MRI for diagnosis. If not, then the true positive cases are more than those recorded.

Author Response

Dear Reviewer.

We are very grateful for your valuable comments that allowed us to systematize our work. Below are our comments with a kind request for a reply whether you find them sufficient or if we should add further explanations.

Ad 1. We have added the value of MRI to the discussion section.

Ad 2. The patient as shown in the photo came to our clinic for treatment, with no prior orthopedic intervention, and no treatment. He had only a history of diabetes, The photo is an illustration of what hip joint deterioration looks like with untreated AVN (right side), and the left side with progressive AVN.

Ad 3. MRI was used in all studies to diagnose AVN. Only in the study by Tse et al. diagnosis was made based on plain film, nuclear scan, or magnetic resonance image. We have added a sufficient paragraph in the discussion. 

Best regards,

Round 2

Reviewer 2 Report

Τhe legend to the figure has been modified. The abbreviations need to be removed.

The value of MRI needs to be included in addition to the introduction (suggested reference PMID: 17555906)

Author Response

Dear Reviewer,

We have modified the figure legend and added MRI value to the introduction.

Best regards